# Solving Schrödinge Bridge problem via stochastic action minimization

## Abstract

The Schrödinger bridge problem is a classical entropy-regularized optimal transport problem that seeks to find optimal diffusion trajectories that transform one probability distribution into another. Although mathematical theory has reached a mature stage, the ongoing research in algorithmic advancements remains a dynamic field, driven by recent innovations in diffusion models. In our research paper, we introduce stochastic Lagrangian and stochastic action as viable alternatives for serving as a direct loss function. We demonstrate the feasibility of incorporating all the vital physical constraints necessary to solve the problem directly into the Lagrangian, providing an intuitive grasp of the loss function and streamlining the training process.

## 1 Introduction

A great deal of ideas in the field of machine learning is inspired by physics and nature. Such transfer of ideas often produces powerful methods that are beautifully simple, intuitive, and theoretically grounded, possibly attributed to the "unreasonable effectiveness of mathematics in natural sciences" (Wigner (1960)). One such example is the Schrödinger bridge problem (SBP). The problem dates back to the beginning of the $20^{th}$ century with the work of Erwin Schrödinger on finding the most likely diffusion evolution of gas particles that transition from one distribution into another (Schrodinger (1931); Schrödinger (1932)). Utilizing classical thermodynamic mathematical apparatus and techniques (probability theory was still under development at that time) Schrödinger derived the diffusion laws for such an evolution. The formulation of the problem as a stochastic generalization of the optimal transport problem (Monge (1781); Kantorovich (1942)) was quickly realized and the name was adopted as an entropy-regularized optimal transport formulation (see for the survey Léonard (2013); Chen et al. (2021b)).

Recently in the field of machine learning, diffusion generative models have emerged as the new state-of-the-art family of deep generative models (Ho et al. (2020); Song et al. (2020); Sohl-Dickstein et al. (2015)). This incentivized further development and interest in the Schrödinger bridge formalism as a generative modeling approach. From the conceptual point of view, both frameworks are rooted in the diffusion evolution. From the technical point of view, the dynamics of both frameworks is guided by the quantity called score $\nabla \log p(x)$ (Hyvärinen & Dayan (2005)). The success of diffusion models is based on the ability of neural networks to learn slight diffusion perturbations in the form of scores from the data and invert the diffusion process, converting one probability distribution into another. While diffusion models learn to convert a data distribution into the tractable probability distribution (usually Gaussian) back and forth, the Schrödinger bridge models are capable of transforming intractable data distributions between each other.

Over the last few years, quite a few modern methods for solving the Schrödinger bridge problem appeared (Chen et al. (2021a); De Bortoli et al. (2021); Shi et al. (2023); Vargas et al. (2021); Tong et al. (2023)) exploiting recent developments in diffusion models (we call them modern SBP methods), demonstrating data dimensionality scalability improvements and new theoretical results, such as likelihood objective for training the Schrödinger bridge model (Chen et al. (2021a)).

The modern numerical solutions of the SBP usually involve two neural networks that build forward and backward trajectories between marginal probability distributions $p_0(x)$ and $p_1(x)$ utilizing Iterative Proportional Fitting Procedure (Ruschendorf (1995); Ireland & Kullback (1968)) to find optimal trajectories.

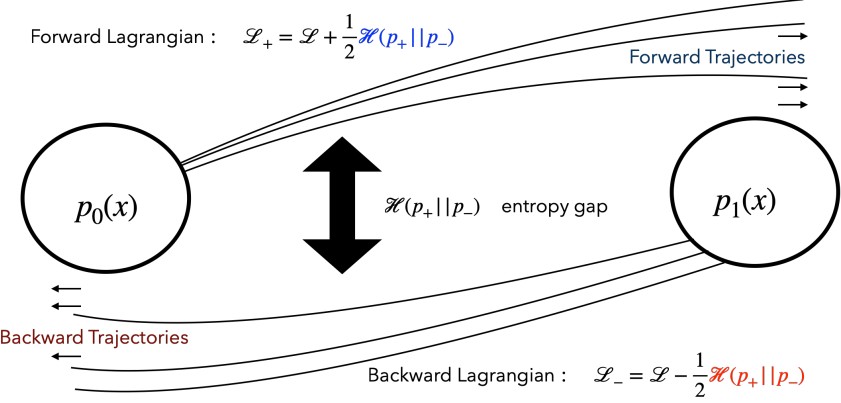

Figure 1: Schematic diagram illustrating entropy gap between forward and backward trajectories and existence for forward and backward Lagrangians taking this gap into account.

As discussed in section (2), SBP is an optimization problem with a constraint on the marginal distribution and evolution that should satisfy the Fokker-Planck equation. To solve this problem, one needs to transform it to the Hamilton-Jacobi dynamic equation and further into the system of forward and backward diffusion processes that can be simulated to find a solution. As demonstrated in (Chen et al. (2021a)) an explicit likelihood loss function exists, but it is not easy to obtain and interpret.

In our paper, we introduce a stochastic Lagrangian and stochastic action. Based on the principle of stochastic Hamilton's least action principle we demonstrate that our Lagrangian can be used to solve the SBP. We demonstrate that all physical constraints can be directly incorporated into the Lagrangian and that our stochastic action can be directly used as a loss function. While the least action principle (one of the most fundamental laws in physics) (Goldstein et al. (2002)) is related to equilibrium, non-stochastic systems that conserve energy, quite a few attempts were made to generalize it to non-equilibrium and stochastic processes (Yasue (1981); Guerra & Morato (1983); Pavon (1995)). Our work is based on the attempts to formulate quantum mechanics in terms of stochastic processes (see an excellent review on the subject (Faris (2014)) originally started by Edward Nelson (Nelson (1966)). While stochastic mechanics fell short in terms of explaining quantum mechanics fully, it was successful at the derivation of the Schrödinger equation and governing stochastic equations. Our stochastic Lagrangian is a combination of the stochastic Lagrangians proposed by Yasue (Yasue (1981)) and Guerra and Morato (Guerra & Morato (1983)) in their quests to derive the Schrödinger equation based on the least action variational principle.

The paper is organized as follows. In section (2) we review the background on the SBP problem. In the following section (3) we introduce the stochastic ßaction. We demonstrate that stochastic action principle leads to meaningful equations of motion and moreover

- Stochastic action minimizes relative entropy between trajectories.
- Stochastic action maximizes Fisher information production.
- Those two conditions are enough to solve the SBP problem
- Stochastic action minimization leads to the Schrödinger equation of motion as a guiding dynamical equation (one can think of it as diffusion equation in the imaginary time).

In section (5) we demonstrate our approach on several experiments. In section (6) we discuss the benefits of our strategy and relations to different approaches to SBP solutions.

## 2 THEORETICAL FOUNDATIONS OF THE DIFFUSION SCHRÖDINGER BRIDGE

The diffusion models in the time continuous scenario can be viewed via the prism of stochastic differential equations (Song et al. (2020)). The main realization is that for every diffusion process described by the stochastic differential equation (SDE)

$$\mathrm{d}x = b_+(x,t)\,\mathrm{d}t + \sqrt{2\beta(t)}\,\mathrm{d}w \tag{1}$$

there exists a diffusion process backward in time described also by the stochastic differential equation

$$\mathrm{d}x = b_-(x, t)\,\mathrm{d}t + \sqrt{2\beta^*(t)}\,\mathrm{d}w \tag{2}$$

Here $b^+$ and $b^-$ are the vector-valued function drift coefficient $b_+(x, t) : \mathbb{R}^d \to \mathbb{R}^d$ and $b_-(x, t) : \mathbb{R}^d \to \mathbb{R}^d$ and $\beta(t)$ and $\beta^*(t)$ are diffusion coefficients that are x-independent by design for simplicity. Here, $\mathrm{d}w$ is the standard $\mathbb{R}^d$ Brownian motion, $\mathrm{d}w$ assumes zero mean $\langle \mathrm{d}w \rangle = 0$ and identity matrix standard deviation $\langle \mathrm{d}w^i \mathrm{d}w^j \rangle = \delta^i_j \, \mathrm{d}t$.

Edward Nelson (Nelson (1966)) was the first to establish a relationship between forward and backward diffusion processes, demonstrating that $\beta^* = \beta$ and uncovering the beautiful relationship between forward drift $b^+$ and backward drift $b^-$ coefficients (3) that paved the way to the formation of the field of stochastic mechanics (Nelson's stochastic mechanics). In machine learning, it now plays a fundamental role in diffusion generative models. See appendix (A.7) for the derivation and discussion.

$$b_- = b_+ - 2\beta \nabla \log p(x) \tag{3}$$

The Schrödinger bridge problem in its dynamic form looks to find optimal trajectories $\mathrm{d}P_+$ that transport one probability distribution $p_0(x)$ into another $p_1(x)$ while minimizing the $\mathbb{KL}-$ divergence between the probability measure induced by an SDE and standard Brownian process $W^\beta$ with a diffusion coefficient $\beta$

$$\begin{aligned} &\min {}_{\mathrm{d}P}\, D_{\mathbb{KL}}(\mathrm{d}P || W^\beta) \\ &x_0 \sim p_0(x), \quad x_N \sim p_1(x) \end{aligned} \tag{4}$$

The theorem 1 allows to recast the problem in the constrained optimization form (8)

**Theorem 1.** *The $\mathbb{KL}-$divergence between two probability measures* $\mathrm{d}P = \rho \mathcal{D}x$ *and* $\mathrm{d}Q = q\mathcal{D}x$ *with two probability densities $\rho$ and $q$ defined on a measure space $\mathcal{D}x = \mathrm{d}x_1 \mathrm{d}x_2 \ldots \mathrm{d}x_n$ induced by two stochastic differential equations*

$$\mathrm{d}x = b\,\mathrm{d}t + \sqrt{2\beta}\,\mathrm{d}w,\ x(0) \sim \pi \tag{5}$$

$$\mathrm{d}x = \gamma\,\mathrm{d}t + \sqrt{2\beta}\,\mathrm{d}w,\ x(0) \sim p_0 \tag{6}$$

*can be decomposed into $\mathbb{KL}-$ divergence between marginal distributions and mean squared error between drift coefficient along the trajectories (Pavon & Wakolbinger (1991))*

$$D_{\mathbb{KL}}(\mathrm{d}P \,||\, \mathrm{d}Q) = D_{\mathbb{KL}}(\pi \,||\, p_0) + \mathbb{E}_{\mathrm{d}P}\Big(\int_0^1 \frac{1}{4\beta}||b - \gamma||^2\Big)\mathrm{d}t \tag{7}$$

*Proof.* This intuitive result is a direct consequence of disintegration theorem and the Girsanov theorem. See appendix (E) for the derivation. ☐

The constrained optimization system (8) has an intuitive interpretation. We seek to minimize the kinetic energy $\frac{1}{2}b_+^2$ such that the dynamics follow a stochastic differential equation with given marginal probability distributions $p_0(x)$ and $p_1(x)$.

$$\begin{aligned} \min_{p,b}\quad & \frac{1}{2}\int\int b_+^2\, p(x, t)\,\mathrm{d}t\,\mathrm{d}x \\ \text{subject to}\quad & \mathrm{d}x = b_+\,\mathrm{d}t + \sqrt{2\beta(t)}\,\mathrm{d}w \quad \text{for}\quad 0 < t < 1 \\ & x_0 \sim p_0(x), \quad x_N \sim p_1(x) \end{aligned} \tag{8}$$

To solve the problem (8), the following system of constraints is reduced to the system of heat equations and decomposition of the probability density into the product of diffusion functions $p(x, t) = \phi(x)\,\hat{\phi}(x)$ via a two-step procedure. First, transforming the problem into the Hamilton-Jacobi equation and then via Hopf-Cole transformation into the system of heat equations (See appendix (I) for the derivation and discussion)

$$\begin{cases} \partial_t \phi = -\beta \Delta \phi \\ \partial_t \hat{\phi} = \beta \Delta \hat{\phi} \end{cases} \text{and}\quad \phi(x, 0)\,\hat{\phi}(x, 0) = p_0(x), \quad \phi(x, 1)\,\hat{\phi}(x, 1) = p_1(x) \tag{9}$$

The following system of equations is usually solved using two neural networks that model optimal policies that build trajectories from probability distribution $p_0(x)$ to $p_1(x)$ and back, and utilize some sort of iterative proportional fitting procedure (Ruschendorf (1995); Ireland & Kullback (1968)) to solve the optimization problem (8).

## 3 Stochastic Action minimization

Forward and backward drift coefficients $b_+$ and $b_-$ can be thought of as mean forward and backward velocities. The time evolution of the probability function is governed by the Fokker-Planck equations

$$\partial_t p = -\nabla \cdot (b_+ p) + \beta \Delta p \qquad (10)$$
$$\partial_t p = -\nabla \cdot (b_- p) - \beta \Delta p \qquad (11)$$

One can introduce the drift and osmotic velocities (Nelson (1966)) defined as

$$v = \frac{b_+ + b_-}{2} \qquad (12)$$

$$u = \frac{b_+ - b_-}{2} = \beta \nabla \log p \qquad (13)$$

The drift velocity $v$ allows to rewrite FPEs equation in the form of the continuity equation

$$\partial_t p + \nabla \cdot (v\, p) = 0 \qquad (14)$$

The objective (8) tends to minimize the mean kinetic energy $\frac{1}{2} b_+^2$ associated with the forward process. However, we have 4 different velocities $b_+, b_-, u, v$ . Interplay of different kinetic energies can form different objectives that can provide additional constraints to forward path kinetic energy minimization. In this section we explore alternative objectives.

### 3.1 Stochastic Action

In classical mechanics the evolution of a physical system with conserved energy corresponds to a stationary point of action (Goldstein et al. (2002))

$$\mathcal{A} = \int_0^1 \mathcal{L}(x(t), \dot{x}(t))\, \mathrm{d}t \qquad (15)$$

where $\mathcal{L}$ is the Lagrangian of the system, $x$ an $\dot{x}$ are the generalized coordinates and velocity. The Lagrangian is usually defined as kinetic energy minus potential energy $\frac{1}{2}|\dot{x}|^2 - V(x)$. Using Euler's fundamental theorem of calculus of variations the Euler-Lagrange equations define equations of motions for the system

$$\frac{d}{\mathrm{d}t}\left(\frac{\partial \mathcal{L}}{\partial \dot{x}}\right) - \frac{\partial \mathcal{L}}{\partial x} = 0 \qquad (16)$$

And from these equations one can recover the Newton's second law $\ddot{x} = -\nabla V$ (we assume $m = 1$).

There were quite a few attempts to introduce a variational approach to stochastic mechanics (Pavon (1995); Yasue (1981); Guerra & Morato (1983)). In stochastic mechanics the stochastic action can be defined as

$$\mathcal{A} = \int p(x) \int_0^1 \mathcal{L}(\dot{x}, x, t) \mathrm{d}x \mathrm{d}t \qquad (17)$$

where the stochastic action (17) differs from the classical action (15) by averaging over spatial coordinates.

For the stochastic action (17), Yasue (Yasue (1981) ) introduced the Lagrangian defined as

$$\mathcal{L}_Y = \frac{u^2 + v^2}{4} = \frac{b_+^2 + b_-^2}{2} \qquad (18)$$

One can immediately see that the following Lagrangian is associated with average kinetic energy of the forward and backward velocities $b^+$ and $b^-$. Lagrangian introduced by Guerra and Morato (Guerra & Morato (1983)) (GM) is defined as

$$\mathcal{L}_{\mathrm{GM}} = \frac{v^2 - u^2}{4} = b_+ b_- \qquad (19)$$

**Lemma:** GM Lagrangian can be rewritten in terms of the $b_+$ and $b_-$ velocities in the alternative form

$$\mathcal{L}_{\text{GM}}^+ = b_+^2 + 2\beta \nabla \cdot b_+ \tag{20}$$

$$\mathcal{L}_{\text{GM}}^- = b_-^2 - 2\beta \nabla \cdot b_- \tag{21}$$

**Proof:**.

$$\int \frac{v^2 - u^2}{4} \, p(x) \, \mathrm{d}x = \int [b_+^2 - 2\beta \, b_+ \nabla \log p] \, p(x) \, \mathrm{d}x = \int [b_+^2 + 2\beta \, \nabla \cdot b_+] \, p(x) \, \mathrm{d}x$$

Analogously,

$$\int \frac{v^2 - u^2}{4} \, p(x) \, \mathrm{d}x = \int [b_-^2 + 2\beta \, b_- \nabla \log p] \, p(x) \, \mathrm{d}x = \int [b_-^2 - 2\beta \, \nabla \cdot b_-] \, p(x) \, \mathrm{d}x$$

We can observe that under the averaging operation of stochastic action Lagrangians (19) and (20) and (21) are identical or $\mathbb{E}[\mathcal{L}_{\text{GM}}] = \mathbb{E}[\mathcal{L}_{\text{GM}}^+] = \mathbb{E}[\mathcal{L}_{\text{GM}}^-]$.

Below, we provide three theorems that form a foundational block for our approach. We start with dynamical equations of motion, associated with the above Lagrangians. Surprisingly, minimizing stochastic actions associated with Yasue and Guerra-Morato Lagrangians lead to the Schödinger equation.

**Theorem 2.** *The stationary points of Yasue and Guerra-Morato actions satisfy the time-dependent Schödinger equation, where the wave function $\psi = \sqrt{p(x)}e^{iS}$ and phase $S$ is related to the drift velocity $v$ via a gradient $v = \nabla S$ (see appendix (F) for the derivation)*

$$i\frac{\partial \psi}{\partial t} = -\beta \Delta \psi \,, \quad \psi(x)\psi^*(x) = p(x) \tag{22}$$

The Schödinger equation can be regarded as a heat equation in imaginary time. Here, the Schödinger wavefunction $\psi$ is complex while the heat equation functions 9 associated with the SBP are real.

Another component that is required for our discussion is the following theorem for the relative entropy between the forward path probability and backward path probability

**Theorem 3.** *Consider the trajectory $x = (x_1, \ldots, x_n)$ and probability density of the trajectory $p(x)$ defined on the measure space $\mathcal{D}x = \mathrm{d}x_1 \mathrm{d}x_2 \cdots \mathrm{d}x_n$. Using the Markov property, the probability density can be decomposed as forward and backward paths*

$$p_+(x) = p(x_1) \, p(x_2|x_1) \, \ldots \, p(x_n|x_{n-1}) \tag{23}$$

$$p_-(x) = p(x_n) \, p(x_{n-1}|x_n) \, \ldots \, p(x_1|x_2) \tag{24}$$

*Then the relative entropy between forward and backward probability densities is*

$$\mathcal{H}(p_+||p_-) = \mathcal{H}_1 - \mathcal{H}_0 + \frac{1}{\beta} \int_0^1 \mathbb{E}[u^2]\mathrm{d}t = \mathcal{H}_1 - \mathcal{H}_0 + \int_0^1 \mathbb{E}[\nabla \cdot (b_+ + b_-)]\mathrm{d}t \tag{25}$$

$$\mathcal{H}(p_-||p_+) = \mathcal{H}_1 - \mathcal{H}_0 - \frac{1}{\beta} \int_0^1 \mathbb{E}[u^2]\mathrm{d}t = \mathcal{H}_0 - \mathcal{H}_1 - \int_0^1 \mathbb{E}[\nabla \cdot (b_+ + b_-)]\mathrm{d}t \tag{26}$$

*where the $\mathcal{H}_0$ and $\mathcal{H}_1$ denote entropies of the marginal probability distributions $p_0$ and $p_1$. One can observe that $\mathcal{H}(p_+||p_-) = -\mathcal{H}(p_-||p_+)$ and due to the non-negative principle of relative entropy this is only possible when*

$$\mathcal{H}(p_+ \,||\, p_-) = 0 \tag{27}$$

$$p_+(x) = p_-(x) \tag{28}$$

$$\int_0^1 \mathbb{E}[v^2] \, \mathrm{d}t = \int_0^1 \mathbb{E}[\nabla \cdot v] \, \mathrm{d}t = H_1 - H_0 = const. \tag{29}$$

*Proof.* See appendix (G) for the derivation. The result of this theorem is that the probability density for the forward trajectories and backward trajectories are the same (Nelson (2020)). □

**Theorem 4.** *Stochastic action minimization of GM Lagrangian (19) is associated Fisher information production (Yang (2021)).*

*Proof.* The new term that appears in GM Lagrangian is $\nabla \cdot b_+$ which is positional Fisher information. Indeed, Fisher information is a way of measuring the amount of information that an observable random variable $x$ carries and is defined as

$$I^{\text{Fisher}}(x) = \int p(x)[\nabla \log p(x)]^2 \mathrm{d}x = \frac{1}{\beta^2}\mathbb{E}[u^2] = -\frac{1}{\beta}\mathbb{E}[\nabla \cdot u] = -\frac{1}{\beta}\mathbb{E}[\nabla \cdot b_+] + \frac{1}{\beta}\mathbb{E}[\nabla \cdot v] \quad (30)$$

Using the fact (29) that $\int_0^1 \mathbb{E}[\nabla \cdot v]\,\mathrm{d}t = \mathcal{H}_1 - \mathcal{H}_0 = \text{const}$, we get the Fisher information production along the trajectory is

$$\mathcal{I}^+ = \int_0^1 I^{\text{Fisher}}(x)\mathrm{d}t = \frac{1}{\beta}(\mathcal{H}_1 - \mathcal{H}_0) - \frac{1}{\beta}\int_0^1 \mathbb{E}[\nabla \cdot b_+]\,\mathrm{d}t \quad (31)$$

$$\mathcal{I}^- = \int_0^1 I^{\text{Fisher}}(x)\mathrm{d}t = \frac{1}{\beta}(\mathcal{H}_0 - \mathcal{H}_1) + \frac{1}{\beta}\int_0^1 \mathbb{E}[\nabla \cdot b_-]\,\mathrm{d}t \quad (32)$$

$\square$

The term $\nabla \cdot b_+$ enters GM Lagrangian (20) and the Fisher information formula (31) with different signs. Analogously, the term $\nabla \cdot b_-$ enters GM Lagrangian (19) and the Fisher information formula (32) also with different signs. We make a conclusion that minimization of stochastic action with GM Lagrangians maximizes Fisher information production along respective trajectories. One can observe that the difference between Fisher Information production directly corresponds to the cross entropy between forward and backward paths.

$$\mathcal{I}^- - \mathcal{I}^+ = \frac{1}{\beta}\int_0^1 \mathbb{E}[\nabla \cdot (b_+ + b_-)]\,\mathrm{d}t = \mathcal{H}(p_+||p_-) + \text{const} \quad (33)$$

## 4 EFFECTIVE LAGRANGIAN

Guerra-Morato (20) and (21) Lagrangians take Fisher information production into account. As we have seen in the previous section the following constraints lead to the Shrödinger equation that transform one marginal probability distribution $p_0(x)$ into another $p_1(x)$. Our goal is to construct policies that predict drift velocities $b_+(x, t)$ and $b_-(x, t)$ as a function of time and position. As we initialize neural networks $b_+$ and $b_-$, these policies are not correlated and hence the relationship between drift velocities (3) is not satisfied and forward and backward probabilities are not equivalent $p_+(x) \neq p_-(x)$. To construct a minimization objective, we construct a Lagrangian that is a combination of Yasue Lagrangian (18), GM Lagrangian (19) and Lagrangian associated with cross entropy (3).

$$\mathcal{L}_{\text{effective}} = \mathcal{L}_{\text{Yasue}} + \mathcal{L}_{\text{GM}} + \beta\,\mathcal{H}(p_+||p_-) = \frac{b_+^2 + b_-^2}{2} + b_+ b_- + \frac{\beta}{2}\nabla \cdot (b_+ + b_-). \quad (34)$$

As we train the policies, the entropy gap shrinks bringing $p_+$ and $p_-$ closer to each other until they merge satisfying the condition for the cross-entropy $\mathcal{H}(p_+||p_-) = 0$. During the training, we utilize the symmetry of the problem. We sample trajectories from one of the networks for $b_+$ or $b_-$ iteratively. We freeze the sampling network. We optimize the effective Lagrangian to fit the trajectories for the active network. We disregard terms that are frozen during the training to obtain the following iterative procedure for the effective Lagrangian.

$$\text{Forward Loss} = \mathbb{E}_{\mathrm{dP}_-}\left[\frac{1}{2}b_+^2 + b_+ b_- + \frac{\beta}{2}\nabla \cdot b_+\right] \quad \text{samples from backward trajectory.} \quad (35)$$

$$\text{Backward Loss} = \mathbb{E}_{\mathrm{dP}_+}\left[\frac{1}{2}b_-^2 + b_+ b_- + \frac{\beta}{2}\nabla \cdot b_-\right] \quad \text{samples from forward trajectory.} \quad (36)$$

The following iterative minimization of the forward and backward loss functions correspond to the iterative proportionate fitting procedure (Ireland & Kullback (1968))

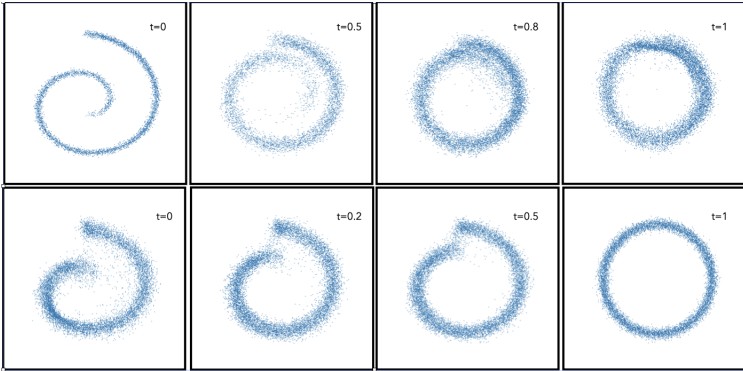

Figure 2: Illustration of the forward process (top row) for the forward trajectory from spiral to circle and backward process (bottom row) for transforming circle back into spiral.

## 5 EXPERIMENTS

### 5.1 TOY MODEL

We demonstrate the utility of our approach on the toy model dataset in 2D (figure 2). We utilize two time-parametrized MLP neural networks that learn to transform the noisy circle dataset into the spiral dataset. Discretization of 100 time datapoints to build a stochastic differential equation with constant noise $\beta(t) = 0.5$. Forward and backward loss functions (35) and (36) were iteratively minimized until convergence.

## 6 DISCUSSION AND CONCLUSIONS

In our work, we introduced stochastic Lagrangian suitable and stochastic action. We use stochastic variational minimization principle to train SB networks. Our approach offers great simplicity since it does not require transformation into the Euler-Lagrange equations. The stochastic Lagrangian incorporates all the necessary physical constraints such as Fisher information and cross-entropy and in our work we demonstrate that this is sufficient to obtain a numerical solution to the SBP. Our approach is based on the congruence of generative diffusion models and SBP formalism as in the works of (Chen et al. (2021a); De Bortoli et al. (2021); Shi et al. (2023)). Our approach is the most similar to the work of (Chen et al. (2021a)) on likelihood training of the SBP. We utilize the same scheme of forward and backward stochastic differential equations and iterative proportionate fitting algorithm to train two neural networks that predict forward and backward drift coefficients. However, the governing dynamical equations are different. In our case, the dynamics follow the Schödinger equation with imaginary part and traditional - diffusion heat equations. Moreover, one can derive the loss function of (Chen et al. (2021a)) by interpreting the stochastic Lagrangians (see appendix (A.4) for the derivation). In conclusion, the stochastic action minimization approach offers a new flexible and interpretable approach to solve SBP via the lense of stochastic mechanics.

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

amsmath

# A APPENDIX

The time evolution of probability density is modeled by the Kolmogorov Forward Equation (KFE), also known as Fokker-Planck equation (FPE) Risken (1996). Two acronyms we use interchangeably.

## A.1 KOLMOGOROV BACKWARD EQUATION

To derive the time evolution operator with respect to initial variables $x_0$ and $t_0$ we use the Chapman-Kolmogorov identity Kallianpur & Sundar (2014)

$$p(x, t \mid x_0, t_0 - \mathrm{d}t_0) = \int p(x, t \mid z, t_0) \, p(z, t_0 \mid x_0, t_0 - \mathrm{d}t_0) \, \mathrm{d}z \tag{37}$$

Denoting $\mathrm{d}x_0 = z - x_0$, we decompose $p(x, t \mid z, t_0)$ around $x_0$ using Taylor expansion

$$p(x, t \mid z, t_0) = p(x, t \mid x_0, t_0) + \frac{\partial}{\partial x_0^i} p(x, t \mid x_0, t_0) \mathrm{d}x_0^i + \frac{1}{2} \frac{\partial^2}{\partial x_0^i \, \partial x_0^j} \, p(x, t \mid x_0, t_0) \, \mathrm{d}x_0^i \, \mathrm{d}x_0^j + O(\mathrm{d}x_0^3) \, .$$

Plugging the decomposition into the equation 37 and using equation 42 with an assumption $\mathrm{d}t^2 = 0$, $\mathrm{d}t \, \mathrm{d}w^i = 0$ and $\mathrm{d}w^i \, \mathrm{d}w^j = \delta_j^i \mathrm{d}t$ we obtain

$$
\begin{aligned}
p(x, t \mid x_0, t_0 - \mathrm{d}t_0) = &\int p(x, t \mid x_0, t_0) \, p(z, t_0 \mid x_0, t_0 - \mathrm{d}t_0) \, \mathrm{d}z \, + \\
& b^i(x_0, t_0) \, \frac{\partial}{\partial x_0^i} p(x, t \mid x_0, t_0) \, \mathrm{d}t_0 \int p(z, t_0 \mid x_0, t_0 - \mathrm{d}t_0) \, \mathrm{d}z \, + \\
& + \beta^{ij}(x_0, t_0) \frac{\partial^2}{\partial x_0^i \, \partial x_0^j} \, p(x, t \mid x_0, t_0) \int p(z, t_0 \mid x_0, t_0 - \mathrm{d}t_0) \, \mathrm{d}z \, + \, O(\mathrm{d}t_0^2)
\end{aligned}
$$

Since $\int p(z, t_0 \mid x_0, t_0 - \mathrm{d}t_0) \, \mathrm{d}z = 1$, the equation takes form

$$
\begin{aligned}
p(x, t \mid x_0, t_0 - \mathrm{d}t_0) - p(x, t \mid x_0, t_0) = \, & b^i(x_0, t_0) \frac{\partial}{\partial x_0^i} p(x, t \mid x_0, t_0) \, \mathrm{d}t_0 \, + \\
& + \beta^{ij}(x_0, t_0) \frac{\partial^2}{\partial x_0^i \, \partial x_0^j} \, p(x, t \mid x_0, t_0) \mathrm{d}t_0 \, + \, O(\mathrm{d}t_0^2)
\end{aligned}
$$

Dividing by $\mathrm{d}t_0$ and taking the limit $\mathrm{d}t_0 \to 0$ we arrive at the Kolmogorov Backward Equation

$$-\frac{\partial}{\partial t_0} [p(x, t \mid x_0, t_0)] = b^i(x_0, t_0) \frac{\partial}{\partial x_0^i} p(x, t \mid x_0, t_0) + \beta^{ij}(x_0, t_0) \frac{\partial^2}{\partial x_0^i \, \partial x_0^j} \, p(x, t \mid x_0, t_0) \, . \tag{38}$$

## A.2 Probability flow ODE for the Kolmogorov Backward Equation

Kolmogorov Backward equation can be compactly rewritten in the following form

$$-\frac{\partial}{\partial t_0}[p(x,t\,|\,x_0,t_0)] = b^i(x_0,t_0)\frac{\partial}{\partial x_0^i}p(x,t\,|\,x_0,t_0)+$$

$$\frac{\partial}{\partial x_0^j}[\beta^{ij}(x_0,t_0)\frac{\partial}{\partial x_0^i}p(x,t\,|\,x_0,t_0)] - \frac{\partial p(x,t\,|\,x_0,t_0)}{\partial x_0^i}\frac{\partial \beta^{ij}(x_0,t_0)}{\partial x_0^j} =$$

$$[b^i(x_0,t_0)-\frac{\partial \beta^{ij}(x_0,t_0)}{\partial x_0^j}]\frac{\partial}{\partial x_0^i}p(x,t\,|\,x_0,t_0) +\frac{\partial}{\partial x_0^i}[\beta^{ij}(x_0,t_0)\frac{\partial}{\partial x_0^j}\log p(x,t\,|\,x_0,t_0)p(x,t\,|\,x_0,t_0)] =$$

If we introduce backward drift coefficients

$$b_{B_1}^i(x_0,t_0) = b^i(x_0,t_0) \ - \ \frac{\partial \beta^{ij}(x_0,t_0)}{\partial x_0^j} \tag{39}$$

and

$$b_{B_2}^i(x_0,t_0) = \beta^{ij}(x_0,t_0)\frac{\partial}{\partial x_0^j}\log p(x,t\,|\,x_0,t_0) \tag{40}$$

the KBE can be written as

$$-\frac{\partial}{\partial t_0}[p(x,t\,|\,x_0,t_0)] = b_{B_1}^i(x_0,t_0)\frac{\partial}{\partial x_0^i}p(x,t\,|\,x_0,t_0) \ + \ \frac{\partial}{\partial x_0^i}[b_{B_2}^i(x_0,t_0)p(x,t\,|\,x_0,t_0)] \tag{41}$$

## A.3 Itô Formula

Let's consider an arbitrary (scalar) function $\phi(x)$ of the Itô process. The Itô differential of $\phi(x,t)$ can be written as Särkkä & Solin (2019):

$$\mathrm{d}\phi = \mathrm{d}t\,\partial_t\phi+\mathrm{d}x^i\,\partial_i\phi+\frac{1}{2}\partial_i\partial_j\phi(x,t)\mathrm{d}x^i\mathrm{d}x^j+O(\mathrm{d}x^3) = \mathrm{d}t\,\partial_t\phi+b^i(x,t)\partial_i\phi\,\mathrm{d}t+\sigma_k^i(x,t)\,\mathrm{d}w^k\partial_i\phi +$$

$$+ \ \beta^{ij}(x,t)\partial_i\partial_j\phi(x,t)\,\mathrm{d}t + O(\mathrm{d}t^2) \tag{42}$$

Taking the expectation with respect to $x$ and dividing both sides by $\mathrm{d}t$ we obtain

$$\frac{d\mathbb{E}[\phi]}{\mathrm{d}t} = \mathbb{E}[\partial_t\phi] + \mathbb{E}[b^i(x,t)\partial_i\phi] + \mathbb{E}[\beta^{ij}(x,t)\partial_i\partial_j\phi(x,t)] \tag{43}$$

## A.4 Application of Itô formula

We can apply Itô formula 42 to derive the SDE for Schrödinger terms $\phi$ and $\hat{\phi}$. For the sampling from the forward trajectory defined by SDE 136

$$d\log\phi = (\partial_t\log\phi + 2\beta\nabla^i\log\phi\nabla_i\log\phi + \beta\Delta\log\phi)dt + \sqrt{2\beta}\nabla\log\phi\,\mathrm{d}w =$$

$$(\frac{1}{\phi}\partial_t\phi + 2\beta|\nabla\log\phi|^2 + \beta\nabla\cdot\frac{\nabla\phi}{\phi})dt + \sqrt{2\beta}\nabla\log\phi\,\mathrm{d}w =$$

$$(\frac{1}{\phi}\partial_t\phi + 2\beta|\nabla\log\phi|^2 + \beta\frac{\Delta\phi}{\phi} - \beta\frac{|\nabla\phi|^2}{\phi^2})dt + \sqrt{2\beta}\nabla\log\phi\,\mathrm{d}w =$$

$$(\frac{1}{\phi}[\partial_t\phi + \beta\Delta\phi] + 2\beta|\nabla\log\phi|^2 - \beta|\nabla\log\phi|^2)dt + \sqrt{2\beta}\nabla\log\phi\,\mathrm{d}w =$$

$$\beta|\nabla\log\phi|^2dt + \sqrt{2\beta}\nabla\log\phi\,\mathrm{d}w$$

Analogously, the evolution for the $\log \hat{\phi}$ when an SDE is defined by the forward process 136

$$d \log \hat{\phi} = (\partial_t \log \hat{\phi} + 2\beta \nabla^i \log \phi \nabla_i \log \hat{\phi} + \beta \Delta \log \hat{\phi}) dt + \sqrt{2\beta} \nabla \log \hat{\phi} \, dw =$$

$$(\frac{1}{\hat{\phi}} \partial_t \hat{\phi} + 2\beta \nabla \log \phi \cdot \nabla \log \hat{\phi} + \beta \nabla \cdot \frac{\nabla \hat{\phi}}{\hat{\phi}}) dt + \sqrt{2\beta} \nabla \log \hat{\phi} \, dw =$$

$$(\frac{1}{\hat{\phi}} \partial_t \hat{\phi} + 2\beta \nabla \log \phi \cdot \nabla \log \hat{\phi} + \beta \frac{\Delta \hat{\phi}}{\hat{\phi}} - \beta \frac{|\nabla \hat{\phi}|^2}{\hat{\phi}^2}) dt + \sqrt{2\beta} \nabla \log \hat{\phi} \, dw =$$

$$(\frac{1}{\hat{\phi}} [\beta \Delta \hat{\phi} + \beta \Delta \hat{\phi}] + 2\beta \nabla \log \phi \cdot \nabla \log \hat{\phi} - \beta |\nabla \log \hat{\phi}|^2) dt + \sqrt{2\beta} \nabla \log \hat{\phi} \, dw =$$

$$(2\beta \nabla \cdot \nabla \log \hat{\phi} + 2\beta \frac{\nabla \hat{\phi} \cdot \nabla \hat{\phi}}{\hat{\phi}^2} + 2\beta \nabla \log \phi \cdot \nabla \log \hat{\phi} - \beta |\nabla \log \hat{\phi}|^2) dt + \sqrt{2\beta} \nabla \log \hat{\phi} \, dw =$$

$$(2\beta \nabla \cdot \nabla \log \hat{\phi} + \beta |\nabla \log \hat{\phi}|^2 + 2\beta \nabla \log \phi \cdot \nabla \log \hat{\phi}) dt + \sqrt{2\beta} \nabla \log \hat{\phi} \, dw$$

## A.5 KOLMOGOROV FORWARD EQUATION

With the help of Itô Formula, we can elegantly derive Kolmogorov forward equation. The expectation of the $\phi$ function can be written in terms of the L2 inner product. L2 inner product between two functions $\phi$ and $p$ is defined as follows:

$$\mathbb{E}[\phi(x)] = \langle \phi, p \rangle = \int \phi(x) \, p(x) \, dx \tag{44}$$

Time evolution of the $\phi$ function can be compactly written in the operator form

$$\frac{d}{dt} \langle \phi, p \rangle = \langle \mathbb{A} \, \phi, p \rangle \tag{45}$$

with an operator $\mathbb{A}$ defined as

$$\mathbb{A} = b^i(x, t) \partial_i + \beta^{ij} \partial_i \partial_j \tag{46}$$

The conjugate operator $\mathbb{A}^\dagger$ for the time evolution for probability density $p$

$$\frac{d}{dt} < \phi, p > = < \mathbb{A} \, \phi, p > = < \phi, \mathbb{A}^\dagger p > \tag{47}$$

can be obtained by using integration by parts $(b^i(x,t)\partial_i)^\dagger = -\partial_i[b^i(x,t)\,(\cdot)]$ and $(\beta^{ij}(x,t)\partial_i\partial_j)^\dagger = \partial_i\partial_j\{\beta^{ij}(x,t)(\cdot)\}$.

The conjugate operator $\mathbb{A}^\dagger$ can be written

$$\mathbb{A}^\dagger = -\partial_i[b^i(x,t)\,(\cdot)] + \partial_i\partial_j\{\beta^{ij}(x,t)(\cdot)\} \tag{48}$$

Since the equation 47 holds for all functions $\phi$, the conjugate equation for probability density should also be true

$$\frac{\partial}{\partial t} p(x,t) = \mathbb{A}^\star p(x,t) \tag{49}$$

We can write down the Kolmogorov Forward Equation in the most general form

$$\frac{\partial}{\partial t}[p(x,t)] = -\partial_i[b^i(x,t)\,p(x,t)] + \partial_i\partial_j\{\beta^{ij}(x,t)\,p(x,t)\}. \tag{50}$$

## A.6 PROBABILITY FLOW ODE FOR THE KOLMOGOROV FORWARD EQUATION

Rewriting the KFE equation 50 as

$$\frac{\partial}{\partial t}[p(x,t)] = -\partial_i[b^i(x,t)p(x,t)] + \partial_i[p(x,t)\,\partial_j\beta^{ij}(x,t)] + \partial_i[\beta^{ij}(x,t)\partial_j p(x,t)]$$

which can be compactly written as

$$\frac{\partial}{\partial t}[p(x,t)] = -\partial_i[b_F^i(x,t)\,p(x,t)] \tag{51}$$

with the new forward drift coefficient defined as

$$b_F^i(x,t) = b^i(x,t) - \partial_j\beta^{ij}(x,t) - \beta^{ij}(x,t)\partial_j\log p(x,t), \tag{52}$$

which corresponds to a standard ODE

$$\mathrm{d}x = b_F(x,t)\mathrm{d}t$$

### A.7 REVERSE TIME STOCHASTIC DIFFERENTIAL EQUATION

In this section, for brevity, the probability at $x_0$ is always referred to at time $t_0$ such that $p(x_0)$ always assumes $p(x_0, t_0)$ and $p(x,t)$ is just abbreviated as $p(x)$. Forward and Backward Kolmogorov equation describe time evolution of probability distribution corresponding to a SDE forward in time as a function of time boundary conditions.

In what follows is a derivation of stochastic differential equation that corresponds to time evolution probability density backward in time or $\partial_{t_0}p(x_0|x_t)$ derived by Brian Anderson (1982). We are interested in deriving a Fokker-Planck equation for probability density function evolving backward in time $\partial_{t_0}p(x|x_0)$.

$$\partial_{t_0}p(x_0,x_t) = p(x_0)\partial_{t_0}\,p(x|x_0)\,+\,p(x|x_0)\,\partial_{t_0}p(x_0) = p(x_0)\,\text{KBE} + p(x|x_0)\,\text{KFE} =$$

First, we perform several algebraic manipulations

$$p(x_0)\,\frac{\partial}{\partial x_0^i}p(x|x_0) = p(x_0)\frac{\partial}{\partial x_0^i}\frac{p(x,x_0)}{p(x_0)} = \frac{p(x_0)\frac{\partial}{\partial x_0^i}p(x,x_0) - p(x,x_0)\frac{\partial}{\partial x_0^i}p(x_0)}{p(x_0)} =$$

$$\frac{\partial}{\partial x_0^i}p(x,x_0) - p(x,x_0)\frac{\partial}{\partial x_0^i}\log p(x_0)$$

Substituting

$$= -(b_{B_1}^i(x_0) + b_{B_2}^i(x_0))[\frac{\partial}{\partial x_0^i}p(x,x_0) - p(x,x_0)\frac{\partial}{\partial x_0^i}\log p(x_0)] -$$

$$- p(x,x_0)\,\frac{\partial}{\partial x_0^i}[b_{B_2}^i(x_0) + b_F^i(x_0)] - p(x,x_0)\,b_F^i(x_0)\frac{\partial}{\partial x_0^i}\log p(x_0) =$$

$$- [b_{B_1}^i(x_0) + b_{B_2}^i(x_0)]\frac{\partial}{\partial x_0^i}p(x,x_0) - p(x,x_0)\frac{\partial}{\partial x_0^i}[b_{B_2}^i(x_0) + b_F^i(x_0)] +$$

$$p(x,x_0)[b_{B_1}^i(x_0) + b_{B_2}^i(x_0) - b_F^i(x_0)]\frac{\partial}{\partial x_0^i}\log p(x_0)$$

Now, looking at the last term

$$p(x,x_0)[b_{B_1}^i(x_0)+b_{B_2}^i(x_0)-b_F^i(x_0)]\frac{\partial}{\partial x_0^i}\log p(x_0) = p(x,x_0)\beta^{ij}(x_0)\frac{\partial}{\partial x_0^j}\log p(x,x_0)\frac{\partial}{\partial x_0^i}\log p(x_0) =$$

$$\beta^{ij}(x_0)\frac{\partial}{\partial x_0^j}p(x,x_0)\frac{\partial}{\partial x_0^i}\log p(x_0) = \beta^{ij}(x_0)\frac{\partial}{\partial x_0^i}p(x,x_0)\frac{\partial}{\partial x_0^j}\log p(x_0)$$

Here, we used $\beta^{ij} = \beta^{ji}$. We get

$$\partial_{t_0}p(x_0,x_t) = -[b_{B_1}^i(x_0) + b_{B_2}^i(x_0) - \beta^{ij}(x_0)\frac{\partial}{\partial x_0^j}\log p(x_0)]\frac{\partial}{\partial x_0^i}p(x,x_0) -$$

$$p(x,x_0)\frac{\partial}{\partial x_0^i}[b_{B_2}^i(x_0) + b_F^i(x_0)]$$

or

$$\partial_{t_0} p(x_0, x_t) = -\frac{\partial}{\partial x_0^i} [(b_{B_2}^i(x_0) + b_F^i(x_0)) \, p(x_0, x_t)] \tag{53}$$

We were able to shape the time evolution of joint probability distribution $p(x_0, x_t)$ into the probability flow representation of the Kolmogorov forward equation with the drift coefficient

$$b_{B_2}^i(x_0) + b_F^i(x_0) = b_F^i(x_0) + \beta^{ij} \frac{\partial}{\partial x_0^j} \log p(x, x_0) - \beta^{ij} \frac{\partial}{\partial x_0^j} \log p(x_0) \,. \tag{54}$$

The above equation can be converted into the standard Kolmogorov forward equation 50

$$\frac{\partial}{\partial t_0} [p(x, x_0)] = -\frac{\partial}{\partial x_0^i} [f^i(x_0, t) \, p(x, x_0)] + \frac{\partial}{\partial x_0^i} \frac{\partial}{\partial x_0^j} \{\beta^{ij}(x_0, t_0) \, p(x, x_0)\}] \,. \tag{55}$$

with the drift coefficient

$$f^i(x_0) = b^i(x_0) - \frac{\partial}{\partial x_0^j} \beta^{ij}(x_0) - 2\beta^{ij} \frac{\partial}{\partial x_0^j} \log p(x_0, t) \,. \tag{56}$$

if the forward diffusion process is $x$-independent $\beta^{ij}(x, t) = \beta^{ij}(t)$ the reverse drift coefficient assumes a well-known form

$$f(x_0) = b(x_0) - 2\beta(t)\nabla \log p(x_0, t) \,. \tag{57}$$

## B  FOKKER-PLANCK DIFFUSION EQUATION

Below, we only consider the special case of a diagonal and position-independent diffusion tensor by setting $\sigma(x, t) = \sqrt{2\beta_t} \, I_{d \times d}$, where $I_{d \times d}$ stands for the $d-$ dimensional identity matrix. As a result, the FPE equation assumes the following form

$$\partial_t \, p_t(x) = -\nabla \cdot [b_t(x) \, p_t(x)] + \beta_t \, \Delta \, p_t(x) \,. \tag{58}$$

Note the appearance of the Laplacian in the FPE equation. Assume that $b_t(x)$ is a gradient of some convex function $\Phi_t(x)$, playing the role of some convex function $b_t(x) = \nabla\Phi_t(x)$ is a potential energy. Then the corresponding SDE:

$$dx_t = \nabla\Phi_t \, dt + \sqrt{2\,\beta_t} \, \mathrm{d}w \tag{59}$$

and the FPE

$$\partial_t p_t = \nabla \cdot (p_t \, \nabla\Phi_t) + \beta_t \Delta \, p_t \tag{60}$$

By direct substitution, one can verify that $p_s(t)$ is a stationary solution of the FPE equation

$$p_s(t) = Z^{-1} \exp(-2\,\beta^{-1} \, \Phi(x)) \,, \quad Z = \int \exp\left(-2\,\beta^{-1} \, \Phi(x)\right) dx \,. \tag{61}$$

The Fokker-Planck equation can also be viewed as the gradient flow for the Dirichlet

## C  SPACE OF PROBABILITY MEASURES

We are considering locally compact, separable, and complete metric spaces and specifically $X = \mathbb{R}^d$. For simplicity, all measures are Borel measures and the space of probability measures is denoted $\mathcal{P}(X)$. Let's consider the functional $\mathcal{F}[\rho] := \mathcal{P}(\mathbb{R}^d)$

$$\mathcal{F}[\rho] = \int_{\mathbb{R}^d} (\rho \log(\rho) + \rho \, V) \, \mathrm{d}x \tag{62}$$

If we make a substitution $\eta := e^V \rho$, one can rewrite the functional as the relative entropy with respect to the measure $e^{-V} \mathrm{d}x$.

$$\mathcal{F}[\rho] = \int_{\mathbb{R}^d} \eta \, \log(\eta) \, e^{-V} \, \mathrm{d}x \tag{63}$$

For a convex function $\eta \log \eta$ we can apply Jensen's inequality to show that the functional $\mathcal{F}$ is positive

$$\mathcal{F}[\rho] = \int_{\mathbb{R}^d} \eta \, \log(\eta) \, e^{-V} \, \mathrm{d}x \geq (\int_{\mathbb{R}^d} \eta \, e^{-V} \, \mathrm{d}x) \, \log(\int_{\mathbb{R}^d} \eta \, e^{-V} \, \mathrm{d}x) =$$
$$= (\int_{\mathbb{R}^d} \rho \, \mathrm{d}x) \, \log(\int_{R^d} \rho \, \mathrm{d}x) = 0 \quad (64)$$

The functional $\mathcal{F}$ reaches its minimum 0 when $\eta = 1$ or when $\rho = e^{-V}$.

## D  NELSON THEORY

In his work, Edward Nelson Nelson (1966) derives the Schrödinger equation via Brownian motion. considering a stochastic Brownian motion

$$dx_t = b_+(x, t)\mathrm{d}t + \sqrt{2\beta_t} \, dw_t \quad (65)$$

Nelson considers the stochastic process evolving backward in time.

$$dx_t = b_-(x, t)\mathrm{d}t + \sqrt{2\beta_t^\star} \, dw_t \quad (66)$$

In his theory, in 1965 Nelson shows that $\beta_t^\star = \beta_t$ and for the first time established the relationship between forward $b_+$ and backward $b_-$ drift coefficients. In the pursuit of establishing the relationship between drift coefficients $b_+$ and $b_-$ he introduces two mathematical constructs: forward and backward derivatives defined as an evolution in time forward and backward

$$\mathcal{D}_+ F(t) = \lim_{\mathrm{d}t \to 0^+} \mathbb{E}_t \frac{F(t + \mathrm{d}t) - F(t)}{\mathrm{d}t} \quad (67)$$

$$\mathcal{D}_- F(t) = \lim_{\mathrm{d}t \to 0^+} \mathbb{E}_t \frac{F(t) - F(t - \mathrm{d}t)}{\mathrm{d}t}, \quad (68)$$

where $\mathbb{E}_t$ stands for the conditional expectation with respect to the present time $t$.

$$\mathbb{E}_t F(t') = \int F(x'(t')) \, p(x', t'|x, t)\mathrm{d}x' \quad (69)$$

where the $p(x', t'|x, t)$ is the conditional probability density. One can see that by the definition the forward and backward velocities are

$$D_+ x(t) = b_+(x, t) \quad (70)$$
$$D_- x(t) = b_-(x, t). \quad (71)$$

According to Itô's lemma **??** for any function $f$

$$df(x, t) = \frac{\partial f}{\partial t}\mathrm{d}t + \mathrm{d}x^i \nabla_i f + \frac{1}{2}\mathrm{d}x^i \mathrm{d}x^j \frac{\partial^2 f}{\partial x_i \partial x_j} + o(\mathrm{d}t) \quad (72)$$

And one can obtain the forward and backward processes in terms of forward and backward derivative operators

$$\mathcal{D}_+ f(x, t) = (\frac{\partial}{\partial t} + b_+^i \nabla_i + \beta \Delta) f(x, t) \quad (73)$$

$$\mathcal{D}_- f(x, t) = (\frac{\partial}{\partial t} + b_-^i \nabla_i - \beta^* \Delta) f(x, t) \quad (74)$$

$$(75)$$

Considering the space-time configuration $M \times I$ such that $\int_{M \times I} f dp = \int_I \mathbb{E}_p f(x(t), t)\mathrm{d}t$ and $\mathbb{E}_p[f] = \int f(x)dp$

Let $f(x)$ and $g(x)$ be two smooth functions with compact support in $\mathbb{C}_0^\infty$ and we define $F(t) = g(x(t))$ and $G(t) = g(x(t))$ be two functions of time.

To establish the relationship between $b_+$ and $b_-$ we need two identities

$$\mathbb{E}_p[F(b)G(b) - F(a)G(a)] = \int_a^b \mathbb{E}_p[\mathcal{D}_+F(t)G(t)]\mathrm{d}t + \int_a^b \mathbb{E}_p[F(t)\mathcal{D}_-G(t)]\mathrm{d}t \qquad (76)$$

To prove this identity, we divide the interval $[a, b]$ into $N$ equal intervals $t_j = a + j\frac{b-a}{N}$ for $j = 0 \ldots N$. Then

$$\mathbb{E}_p[F(b)G(b) - F(a)G(a)] = \lim_{N\to\infty} \sum_{j=1}^{N-1} \mathbb{E}_p[F(t_{j+1})G(t_j) - F(t_j)G(t_{j-1})] =$$

$$\lim_{N\to\infty} \sum_{j=1}^{N-1} \mathbb{E}_p[(F(t_{j+1}) - F(t_j))\frac{G(t_j) + G(t_{j-1})}{2} + \frac{F(t_{j+1}) + F(t_j)}{2}(G(t_j) - G(t_{j-1}))] =$$

$$\lim_{N\to\infty} \sum_{j=1}^{N-1} \mathbb{E}_p[(\mathcal{D}_+F(t_j))G(t_j) + F(t_j)\mathcal{D}_-G(t_j)]\frac{b-1}{N} =$$

$$\int_a^b \mathbb{E}_p[(\mathcal{D}_+F(t))G(t) + F(t)\mathcal{D}_-G(t)]\mathrm{d}t\,.$$

On the other hand,

$$\mathbb{E}_p[F(b)G(b) - F(a)G(a)] = \int_a^b \mathbb{E}_p d(F(t)G(t))\mathrm{d}t = \qquad (77)$$

$$= \int_a^b \mathbb{E}_p dF(t)G(t)\mathrm{d}t + \int_a^b \mathbb{E}_p F(t)dG(t)\mathrm{d}t + \int_a^b \mathbb{E}_p dF(t)dG(t)\mathrm{d}t =$$

$$= \int_a^b \mathbb{E}_p \mathcal{D}_+F(t)G(t)\mathrm{d}t + \int_a^b \mathbb{E}_p F(t)\mathcal{D}_+G(t)\mathrm{d}t + \int_a^b \mathbb{E}_p 2\beta^{ij}\nabla_i f(x)\nabla_j g(x)\mathrm{d}t$$

Eliminating $\mathbb{E}_p[F(b)G(b) - F(a)G(a)]$ we get

$$\mathbb{E}_p f(x,t)\mathcal{D}_-g(x,t) = \mathbb{E}_p f(x,t)\mathcal{D}_+g(x,t) + \mathbb{E}_p 2\beta^{ij}\nabla_i f(x)\nabla_j g(x) = \qquad (78)$$

$$\mathbb{E}_p f(x,t)\mathcal{D}_+g(x,t) - \int f(x)2\beta^{ij}\nabla_i\nabla_j g(x)p(x)\mathrm{d}x - \int f(x)2\beta^{ij}\nabla_j g(x)\frac{\nabla_i p(x)}{p(x)}p(x)\mathrm{d}x \quad (79)$$

Substituting the definitions for $\mathcal{D}_+$ and $\mathcal{D}_-$ we obtain

$$\mathcal{D}_- = \mathcal{D}_+ - 2\beta^{ij}\nabla_i\nabla_j - 2\beta^{ij}\nabla_i \log p(x)\nabla_j. \qquad (80)$$

We immediately recover $\beta^\star = \beta$ and $b_- = b_+ - 2\beta^{ij}\nabla_i \log p(x)$ in the special case, when $\beta^{ij} = \beta\delta(i,j)$ we obtain $b_- = b_+ - 2\beta\nabla \log p(x)$.

For the forward and backward in-time diffusion equations the corresponding Fokker-Planck equations are

$$\partial_t p(x,t) = -\nabla_i(b_+^i p(x,t)) + \beta\Delta p(x,t) \qquad (81)$$

$$\partial_t p(x,t) = -\nabla_i(b_-^i p(x,t)) - \beta\Delta p(x,t) \qquad (82)$$

Nelson introduces the drift and osmotic velocities defined as

$$v^i = \frac{b_+^i + b_-^i}{2} \qquad (83)$$

$$u^i = \frac{b_+^i - b_-^i}{2} = \beta\nabla_i \log p \qquad (84)$$

The drift velocity $\mathbf{v}$ allows to rewrite FPEs equation in the form of the continuity equation

$$\partial_t p + \nabla_i(v^i p) = 0 \qquad (85)$$

One can rewrite the FPEs given in terms of probability density $p$ and drift coefficients $b_+$ and $b_-$ in terms of two dynamical equations of motion for defined velocities $u$ and $v$. For this reason, one needs to introduce the stochastic acceleration

$$a^i = \frac{1}{2}(\mathcal{D}_+ b_-^i + \mathcal{D}_- b_+^i) \tag{86}$$

Using equations of motion for the drift coefficients

$$\mathcal{D}_+ b_-^i = \frac{\partial b_-^i}{\partial t} + (b_+^j \nabla_j)b_-^i + \beta \Delta b_-^i$$

$$\mathcal{D}_- b_+^i = \frac{\partial b_+^i}{\partial t} + (b_-^j \nabla_j)b_+^i + \beta \Delta b_+^i \tag{87}$$

For the external potential $\phi$, Newton's second law reads $\mathbf{F} = m\mathbf{a} = -\nabla\phi$, which allows us to rewrite the diffusion laws of motion in terms of Nelson's equations

$$\frac{\partial v^i}{\partial t} = -\frac{1}{m}\nabla^i\phi - (v^j \nabla_j)v^i + (u^j \nabla_j)u^i + \beta\Delta u^i$$

$$\frac{\partial u^i}{\partial t} = -\beta\Delta v^i - \nabla^i(v^j u_j) \tag{88}$$

The first equation is obtained by summing the equations for the drift coefficients. To obtain the second equation of motion we use the continuity equation by dividing it by $1/p$ and taking gradients on both sides. These two equations give the complete description of the dynamics of a Brownian particle in the context of stochastic mechanics.

### D.1 TIME-INDEPENDENT SCHÖDINGER EQUATION

Nelson equations 88 are nonlinear in nature. However, one can reduce them to linear via a substitution reminiscent of the Hopf-Cole transformation Eberhard (1950); Cole (1951). By definition, the osmotic velocity $u = \beta\nabla\log p$ is a gradient of a function $R = \frac{1}{2}\log p$. We make a similar assumption on the drift velocity $v = 2\beta\nabla S$ being a gradient of some function $S$.

$$u = 2\beta\,\nabla R$$
$$v = 2\beta\,\nabla S \tag{89}$$

Introducing the following complex function $\psi$

$$\psi = e^{R+iS} \tag{90}$$

one can see that $|\psi|^2 = e^{2R} = p$ , which is equivalent to Born's rule. In this case, it is derived rather than postulated. We want to demonstrate that the function $\psi$ satisfies the Schrödinger equation

$$i\hbar\frac{\partial\psi}{\partial t} = (-\frac{\hbar^2}{2m}\Delta + \phi)\psi \tag{91}$$

Indeed, substituting the definition of $\psi$, we need to prove the following equation

$$(\frac{\partial R}{\partial t} + i\frac{\partial S}{\partial t})\psi = i\frac{\hbar}{2m}(\Delta R + i\Delta S + |\nabla R + i\nabla S|^2)\psi - i\frac{1}{\hbar}\phi\,\psi \tag{92}$$

Dividing by $\psi$ and separating real and imaginary parts we obtain two equations:

$$\frac{\partial R}{\partial t} = -\frac{\hbar}{2m}\Delta S - \frac{2\hbar}{2m}\nabla R \cdot \nabla S \tag{93}$$

$$\frac{\partial S}{\partial t} = \frac{\hbar}{2m}\Delta R + \frac{\hbar}{2m}(|\nabla R|^2 + |\nabla S|^2) - \frac{1}{\hbar}\phi \tag{94}$$

Taking the gradients of both equations, we can make a substitution for $\nabla R$ and $\nabla S$ from 89. In addition, we connect the diffusion coefficient $\beta$ to Planck's constant $\beta = \frac{\hbar}{2m}$, and we rewrite the Schrödinger equation in terms of Nelson's velocities $u$ and $v$, given by equation 88 which we already proved.

# E  $\mathbb{KL}-$DIVERGENCE BETWEEN TWO PROBABILITY MEASURES AND GIRSANOV THEOREM

To find the $\mathbb{KL}-$divergence $D_{\mathbb{KL}}(dP||dQ)$ between the probability measures $dP$ and $dQ$ one can decompose the probability densities using Markov properties

$$p(x_1, \dots x_n) = p(x_1)\, p(x_2|x_1) \dots p(x_n|x_{n-1}) \tag{95}$$
$$q(x_1, \dots x_n) = q(x_1)\, q(x_2|x_1) \dots q(x_n|x_{n-1}) \tag{96}$$
$$\tag{97}$$

Using sde evolution for one step $x_{i+1} = x_i + b\Delta t + \sqrt{2\beta}\, dw$ and $x_{i+1} = x_i + \gamma\Delta t + \sqrt{2\beta}\, dw$ with $dw \sim \mathcal{N}(0, \Delta t)$for two SDEs correspondingly, obtain probability for a single step transition in the form of Gaussian distribution

$$p(x_{i+1}|x_i) = \sqrt{\frac{1}{4\pi\beta\Delta t}}\, \exp\frac{(-(x_{i+1} - x_i - b\Delta t)^2}{4\beta\Delta t}) \tag{98}$$
$$q(x_{i+1}|x_i) = \sqrt{\frac{1}{4\pi\beta\Delta t}}\, \exp\frac{(-(x_{i+1} - x_i - b\Delta t)^2}{4\beta\Delta t}) \tag{99}$$

As a result,

$$
\begin{aligned}
D_{\mathbb{KL}}(dP||dQ) &= \int dP\, \ln\frac{dP}{dQ} = \int p(x)\ln\frac{p(x)}{q(x)}\mathcal{D}x = \\
&= D_{\mathbb{KL}}(p(x_0)||q(x_0)) + \sum_i \int p(x)\ln\frac{p(x_i|x_{i-1})}{q(x_i|x_{i-1})}\mathcal{D}x = \\
&= D_{\mathbb{KL}}(p(x_0)||q(x_0)) + \int p(x)\frac{2\mathrm{d}x(\gamma - b)\Delta t + (b^2 - \gamma^2)\Delta t^2}{4\beta\Delta t}\mathcal{D}x = \\
&= D_{\mathbb{KL}}(p(x_0)||q(x_0)) + \frac{1}{4\beta}\int p(x)(2(\gamma - b)\mathrm{d}x + (b^2 - \gamma^2)\Delta t)\,\mathcal{D}x = \\
&= D_{\mathbb{KL}}(p(x_0)||q(x_0)) + \frac{1}{4\beta}\int p(x)||b - \gamma||^2\Delta t\,\mathcal{D}x = \\
&= D_{\mathbb{KL}}(p(x_0)||q(x_0)) + \frac{1}{4\beta}\mathbb{E}_{dP}\int_0^1 ||b - \gamma||^2\Delta t \tag{100}
\end{aligned}
$$

Which completes our proof.

# F  DERIVATION OF SCHRÖDINGER EQUATION IS CALCULUS OF VARIATIONS

The differential of the stochastic action 17

$$\delta A = \frac{1}{4}\int_0^1\int \left(2vp\,\delta v - 2up\delta u + v^2\delta p - u^2\delta p\right)\mathrm{d}x\,\mathrm{d}t \tag{101}$$

Using the equation for continuity 14 and 13 we have the following additional equations for differentials

$$\beta\nabla\delta p = p\delta u + u\delta p \tag{102}$$
$$\partial_t\delta p = \nabla\cdot(p\delta v + v\delta p) \tag{103}$$

Inserting these equations into differential for stochastic action

$$\delta A = \frac{1}{4}\int \left(2vp\,\delta v - 2up\delta u + v^2\delta p - u^2\delta p\right)\mathrm{d}x\,\mathrm{d}t \tag{104}$$

derivation and discussion D.

## G   RELATIVE ENTROPY OF THE FORWARD AND BACKWARD PATH

The probability density of the trajectory $(x_1, x_2, \ldots x_n)$ can be decomposed into the probability of the forward and backward trajectory using Markov property assumption

$$p(x_1, \ldots x_n) = p(x_1)\, p(x_2|x_1) \ldots p(x_n|x_{n-1}) \tag{105}$$

$$p(x_1, \ldots x_n) = p(x_n)\, p(x_{n-1}|x_n) \ldots p(x_1|x_2) \tag{106}$$

As a result, the relative entropy between forward and backward paths can be written

$$H(p_+||p_-) = \int p_+ \ln \frac{p_+}{p_-} \mathcal{D}x = \int p_+ \, ln \frac{p(x_1)}{p_1(x_n)} \mathcal{D}x + \sum_{i=1}^{i=n} \int p_+ \ln \frac{p(x_{i+1}|x_i)}{p(x_i|x_{i+1})} \mathcal{D}x =$$

$$\int p_+ \ln p(x_1)\mathcal{D}x - \int p_+ \ln p(x_n)\mathcal{D}x + \sum_{i=1}^{i=n} \int p_+ \ln \frac{p(x_{i+1}|x_i)}{p(x_i|x_{i+1})} \mathcal{D}x \tag{107}$$

Making an observation that $\int p(x_{i+1}|x_i)dx_{i+1}\mathcal{D}x = 1$ the first term becomes the negative entropy of the marginal distribution at the initial time step.

$$\int p_+ \ln p(x_1)\mathcal{D}x = \int p(x_1) \ln p(x_1)dx_1 = -\mathcal{H}(x_1) \tag{108}$$

The second term is the entropy at the final time step. Indeed,

$$\int p_+ \ln p(x_n)\mathcal{D}x = \int p(x_{n-1})p(x_n|x_{n-1}) \ln p(x_n)dx_{n-1}\mathrm{d}x_n = -\mathcal{H}(x_n) \tag{109}$$

As for the last term, using the Bayes theorem $\frac{p(x_{i+1}|x_i)}{p(x_i|x_{i+1})} = \frac{p(x_{i+1})}{p(x_i)}$

$$\sum_{i=1}^{i=n} \int p_+ \ln \frac{p(x_{i+1}|x_i)}{p(x_i|x_{i+1})}\mathcal{D}x = \sum_{i=1}^{i=n} \int p_+ \ln p(x_{i+1})\mathcal{D}x - \sum_{i=1}^{i=n} \int p_+ \ln p(x_i)\mathcal{D}x =$$

$$\sum_{i=1}^{i=n} \int p(x_i)\mathrm{d}x_i[\int p(x_{i+1}|x_i) \ln p(x_{i+1})\mathrm{d}x_{i+1} - \ln p(x_i)]$$

$$= \sum_{i=1}^{i=n} \mathbb{E}[\int p(x_{i+1}|x_i) \ln p(x_{i+1})\mathrm{d}x_{i+1} - \ln p(x_i)] = \sum_{i=1}^{i=n} \mathbb{E}[\int p(x_{i+1}|x_i) \ln p(x_{i+1})\mathrm{d}x_{i+1} - \ln p(x_i)] =$$

$$\int_{i=1}^{n-1} \mathbb{E}[\frac{\mathbb{E}_t[\ln p(\xi_{i+1})] - \ln p(\xi_i)}{\Delta t}]\Delta t = \int_0^1 \mathbb{E}[\mathcal{D}_+ \ln p(\xi(t))]\mathrm{d}t \tag{110}$$

If we recall the expansion for the forward derivative $\mathcal{D}_+$ from the formula 75 the relative entropy can be rewritten as

$$H(p_+||p_-) = H(x_n) - H(x_1) + \int_0^1 \mathbb{E}[\frac{\partial}{\partial t} + b_+^i \nabla_i + \beta\Delta \ln p]\mathrm{d}t \tag{111}$$

Since $\int_0^1 \mathbb{E}\frac{\partial}{\partial t} \ln p\mathrm{d}t = \int \mathrm{d}x_1 p(x_1) - \int \mathrm{d}x_n p(x_n) = 0$ and recalling the definition of the osmotic velocity 84 we can write

$$\int_0^1 \mathbb{E}[\mathcal{D}_+ \ln p(\xi(t))]\mathrm{d}t = \int_0^1 \mathbb{E}[\nabla_i(u^i - b_+^i)]\mathrm{d}t = \int_0^1 \mathbb{E}[\nabla_i v^i]\mathrm{d}t \tag{112}$$

All together,

$$H(p_+||p_-) = H(x_n) - H(x_1) - \int_0^1 \mathbb{E}[\nabla_i v^i]\mathrm{d}t \tag{113}$$

Analogously, one can prove that

$$H(p_-||p_+) = H(x_1) - H(x_n) + \int_0^1 \mathbb{E}[\nabla_i v^i]\mathrm{d}t \tag{114}$$

we observe that $H(p_+||p_-) = -H(p_-||p_+)$. But since the relative entropy is always nonnegative, this is only possible when $H(p_+||p_-) = 0$.

# H  HAMILTON-JACOBI EQUATIONS AND LINEARIZATION VIA HOPF-COLE TRANSFORMATION

The minimization problem

$$\inf_{p,b} \int_0^1 \int \frac{1}{2} b^i b_i \, p(x,t) \mathrm{d}x \mathrm{d}t \tag{115}$$

$$\frac{dp}{\mathrm{d}t} + \nabla \cdot (pb) = \beta \Delta p \tag{116}$$

$$p(x,0) = p_0(x), \quad p(x,1) = p_1(x) \tag{117}$$

The corresponding Lagrangian is written as

$$\int_0^1 \int \{\frac{1}{2} b^i b_i \, p(x,t) + \psi(x,t) \times (\frac{dp}{\mathrm{d}t} + \nabla \cdot (pb) - \beta \Delta p)\} \, \mathrm{d}x \mathrm{d}t \tag{118}$$

where $\psi(x,t)$ plays the role of Lagrange multiplier. Getting rid of derivatives of probability function $p$ via the integration by part, we can rewrite the above equation as

$$\int_0^1 \int \{\frac{1}{2} b^i b_i \, p(x,t) + (-\frac{d\psi(x,t)}{\mathrm{d}t} - b \cdot \nabla \psi + \beta \Delta \psi)\} \, p(x,t) \, \mathrm{d}x \mathrm{d}t \tag{119}$$

Taking the derivative with respect to $b$ we obtain the expression for the optimal drift.

$$b^{\text{optimal}} = \nabla \psi \tag{120}$$

Substituting this expression back into the equation and make the assumption that the above equation should hold true for all functions $p(x,t)$ we obtain the following Hamilton-Jacobi equation

$$\frac{d\psi(x,t)}{\mathrm{d}t} + \frac{1}{2}||\nabla \psi||^2 = -\beta \Delta \psi \tag{121}$$

which is a Hamiltion-Jacobi equation. This equation can be linearized via Hopf-Cole transformation Eberhard (1950) by making a substitution $\psi = 2\beta \log \phi$. By direct substitution we get

$$\frac{d\phi}{dt} = -\beta \Delta \phi \tag{122}$$

If we look for a decomposition in the form of $p = \phi \hat{\phi}$ and substituting the decomposition into the Fokker-Planck equation we get

$$\hat{\phi}(\frac{d\phi}{dt} + \beta \Delta \phi) + \phi(\frac{d\hat{\phi}}{dt} - \beta \Delta \phi) = 0 \tag{123}$$

which can only be satisfied if and only if

$$\frac{d\hat{\phi}}{dt} = \beta \Delta \hat{\phi} \tag{124}$$

with the transformation $\hat{\phi} = p\phi^{-1} = p \exp \frac{-\psi}{2\beta}$.

# I  CLASSICAL SCHRÖDINGER BRIDGE FORMALISM

**Theorem 5.** *The $\mathbb{KL}-$divergence between two probability measures $\mathrm{d}P = \rho \mathcal{D}x$ and $\mathrm{d}Q = q\mathcal{D}x$ with two probability densities $\rho$ and $q$ defined on a measure space $\mathcal{D}x = \mathrm{d}x_1 \mathrm{d}x_2 \ldots \mathrm{d}x_n$ induced by two stochastic differential equations*

$$\mathrm{d}x = b\,\mathrm{d}t + \sqrt{2\beta}\,\mathrm{d}w, \; x(0) \sim \pi \tag{125}$$

$$\mathrm{d}x = \gamma\,\mathrm{d}t + \sqrt{2\beta}\,\mathrm{d}w, \; x(0) \sim p_0 \tag{126}$$

*can be decomposed into $\mathbb{KL}-$ divergence between marginal distributions and mean squared error between drift coefficient along the trajectories Pavon & Wakolbinger (1991)*

$$D_{\mathbb{KL}}(\mathrm{d}P \,||\, \mathrm{d}Q) = D_{\mathbb{KL}}(\pi \,||\, p_0) + \mathbb{E}_{\mathrm{d}P}(\int_0^1 \frac{1}{4\beta}||b - \gamma||^2)\mathrm{d}t \tag{127}$$

*Proof.* This intuitive result is a direct consequence of disintegration theorem and Girsanov theorem. See appendix E for the derivation. □

The above theorem allows to formulate Schrödinger Bridge problem as finding minimizing the $\mathbb{KL}-$divergence between probability measure induced by an SDE and Brownian motion process

$$\min D_{\mathbb{KL}}(\mathrm{d}P||W_\beta) = \min b^i b_i \tag{128}$$

The Schrödinger Bridge formulation is given as

$$\begin{aligned}
\min_{p,b} \quad & \frac{1}{2}\int\int|b_+|^2\, p(x,t)\mathrm{d}t\mathrm{d}x \\
\text{subject to} \quad & \mathrm{d}x = b_+\,\mathrm{d}t + \sqrt{2\beta(t)}\,\mathrm{d}w \quad \text{for} \quad 0 < t < 1 \\
& p_0(x) = p_0, \quad p_1(x) = p_1(x)e
\end{aligned} \tag{129}$$

If we define a Largrangian

$$\mathcal{L} = \frac{1}{2}\mathbb{E}_{\mathrm{d}P_+}\, b^i_+ b_{+i} \tag{130}$$

the variational approach with Lagrangian constraint can be written as the following minimization problem

$$\min \int_0^1 \mathcal{L}\mathrm{d}t + \psi\int_0^1\mathbb{E}_{\mathrm{d}P}[\frac{\partial p}{\partial t} + \nabla\cdot(pb) - \beta\Delta p]\mathrm{d}t \tag{131}$$

Minimizing with respect to Lagrangian multiplier $\psi$ and $b_+$ gives the well-known Hamilton-Jacobi dynamical equation of motion in tandem with Fokker-Planck equation

$$\frac{d\psi(x,t)}{\mathrm{d}t} + \frac{1}{2}||\nabla\psi||^2 = -\beta\Delta\psi \tag{132}$$

$$\frac{dp}{\mathrm{d}t} + \nabla\cdot(p\nabla\psi) = \beta\Delta p \tag{133}$$

where the optimal value of $b$ is given by

$$b^{\mathrm{optimal}}_+ = \nabla\psi \tag{134}$$

The above equation can be easily transformed into the following forward-backward differential heat equations via the transformation $\phi = \exp\frac{\psi}{2\beta}$ and $\hat{\phi} = p\exp-\frac{\psi}{2\beta}$ and probability density $p(x,t) = \phi(x,t)\hat{\phi}(x,t)$. (See appendix H ) for the derivation.

$$\begin{cases} \frac{d\phi}{\mathrm{d}t} = -\beta\Delta\phi \\ \frac{d\hat{\phi}}{\mathrm{d}t} = \beta\Delta\hat{\phi} \end{cases} \text{and} \quad \phi(x,0)\hat{\phi}(x,0) = p_0(x), \quad \phi(x,1)\hat{\phi}(x,1) = p_1(x) \tag{135}$$

And the optimal drift can be written in terms of $\phi$ and $\hat{\phi}$ functions as $b^{\mathrm{optimal}}_+ = 2\beta\log\phi$ and $b^{\mathrm{optimal}}_+ = -2\beta\log\hat{\phi}$ correspondingly. If the forward trajectory is guided by the drift coefficient $b^{\mathrm{optimal}}_+ = 2\beta\nabla\log\phi$ then the optimal backward trajectory has an optimal drift $b^{\mathrm{optimal}}_- = b^{\mathrm{optimal}}_+ - 2\beta(\nabla\log\phi + \nabla\log\hat{\phi}) = -2\beta\nabla\log\hat{\phi}$. Solution to the optimization problem 8 can be expressed as a path measure simulated by the forward and backward SDEs

$$\mathrm{d}x^i = 2\beta\nabla\log\phi\,\mathrm{d}t + \sqrt{2\beta(t)}\mathrm{d}w^i \quad x_0\sim p_0 \tag{136}$$

$$\mathrm{d}x^i = -2\beta\nabla\log\hat{\phi}\,\mathrm{d}t + \sqrt{2\beta(t)}\mathrm{d}w^i \quad x_0\sim p_1 \tag{137}$$

Using Itó lemma we can write the down SDE equations for $\log\phi$ and $\log\hat{\phi}$ for the forward trajectories 136 (See appendix A.4 for the derivation)

$$\mathrm{d}x = 2\beta\nabla\log\phi\,\mathrm{d}t + \sqrt{2\beta(t)}\mathrm{d}w \tag{138}$$

$$\mathrm{d}\log\phi = \beta|\nabla\log\phi|^2\mathrm{d}t + \sqrt{2\beta}\nabla\log\phi\,\mathrm{d}w \tag{139}$$

$$\mathrm{d}\log\hat{\phi} = (2\beta\nabla\cdot\nabla\log\hat{\phi} + \beta|\nabla\log\hat{\phi}|^2 + 2\beta\nabla\log\phi\cdot\nabla\log\hat{\phi})\mathrm{d}t + \sqrt{2\beta}\nabla\log\hat{\phi}\,\mathrm{d}w \tag{140}$$

Interestingly enough, since $\log p = \log \phi + \log \hat{\phi}$

$$\mathrm{d}x = 2\beta\nabla\log\phi\,\mathrm{d}t + \sqrt{2\beta(t)}\mathrm{d}w \tag{141}$$

$$\mathrm{d}\log p = (2\beta\nabla\cdot\nabla\log\hat{\phi} + \beta|\nabla\log p|^2\mathrm{d}t + \sqrt{2\beta}\nabla\log p\,\mathrm{d}w \tag{142}$$

If one integrates over the trajectories in time, one can immediately recover the loglikelihood of the data point $x_0$ as

$$\log p(x_0) = \mathbb{E}_{\mathrm{d}P_+}[\log p_T(x_N)] - \int_0^1 \mathbb{E}_{\mathrm{d}P_+}(2\beta\nabla\cdot\nabla\log\hat{\phi} + \beta|\nabla\log p|^2)\mathrm{d}t \tag{143}$$

where the expectations are taken over the forward trajectories $\mathrm{d}P_+$.

## J   SCHRÖDINGER BRIDGE FORMALISM AS AN OPTIMAL TRANSPORT

For any two Borel probability measures $b$ and $\nu$ on two Polish space $(\mathcal{X}, d_{\mathcal{X}})$ and $(\mathcal{Y}, d_{\mathcal{Y}})$ and a positive semi-continuous cost function c, $\mathcal{X} \times \mathcal{Y} \to \mathbb{R}^+$, the problem of optimal transport is concerned with finding a solution to the following optimization problem

$$W_p(x,y) := \inf_{\pi\in\Pi(x,y)} \int_{\mathbb{R}^d\times\mathbb{R}^d} c(x,y)\,d\pi(x,y)\,, \tag{144}$$

where $\Pi(b,\nu)$ is a set of measures on $\mathcal{X} \times \mathcal{Y}$ with marginals $b$ and $\nu$. In the case when the role of the cost function plays Euclidean distance $c(x,y) = ||x-y||^p$ with $p \geq 1$, the the $L^p$-Wasserstein distance is introduced

$$W_p(x,y) := \big(\inf_{\pi\in\Pi(x,y)} \int_{\mathbb{R}^d\times\mathbb{R}^d} d(x,y)^p\,d\pi(x,y)\big)^{\frac{1}{p}} \tag{145}$$

This is known as the Kantorovich formulation of optimal transport. When $b$ is absolutely continuous with respect to the Lebesgue measure (i.e. when $b$ has a density), the above can be written in terms of Monge formulation $T_\# b = \nu$ i.f.f. for all Borel sets $A$, $\nu(T(A)) = b(A)$.

Solving the above equation is problematic,

$$\inf_{\pi\in\Pi(b,\nu)} \int_{X\times Y} c(x,y)\,d\pi(x,y) + \epsilon\mathcal{D}\cdot(\pi \mid b\times\nu)\,, \tag{146}$$

where

$$\mathcal{D}_{KL}(p \mid q) = \int_{\mathcal{X}\times\mathcal{Y}} \log\frac{dp}{dq}dp$$

stands for the Kullback-Leibler (KL) divergence between two distributions, p and q. Introducing Gibbs measure $\mathcal{K}$

$$d\mathcal{K}(x,y) = \exp\left(-\frac{c(x,y)}{\epsilon}\right)db(x)\,d\nu(y)\,. \tag{147}$$

one may rewrite the entropy regularized OT problem as

$$\inf_{\pi\in\Pi(b,\nu)} D_{KL}(\pi \mid \mathcal{K})\,. \tag{148}$$

If one replaces Gibbs measure with Wiener measure $\mathcal{W}^\gamma$, one arrives at the static formulation of the Schrödinger Bridge problem

$$\inf_{\pi\in\Pi(b,\nu)} D_{KL}(\pi \mid \mathcal{W}^\gamma)\,. \tag{149}$$

The Schrödinger bridge problem can equivalently be formulated using a dynamic formalism

$$\begin{aligned} \min_{\rho,b} \int_0^1 \int_{\mathbb{R}^d} |\nabla\Phi_t(x)|^2\rho_t(x)\,\mathrm{d}x\,\mathrm{d}t \\ \text{Subject to:} \\ \partial_t\rho_t + \mathrm{div}(\rho_t\nabla\Phi_t) = \beta\Delta\rho_t \quad \text{for} \quad 0 < t < 1, \\ \rho_0 = b, \qquad \rho_1 = \nu. \end{aligned} \tag{150}$$

Now we focus on the Lagrangian point of view which is based on variational principles: Given a Lagrangian $\mathcal{L}(\dot{q}, q, t)$ an object be it a particle or a ray of light) chooses the trajectory $q(t)$ that makes the action $\mathcal{S}$

$$\mathcal{S} = \int_0^1 \mathcal{L}(\dot{x}, x, t) \mathrm{d}t \tag{151}$$

stationary. The action depends on the endpoints $(t_0 = 0, x_0)$ and $(t_1 = 1, x_1)$ and the trajectory $x(t)$ must obey the Euler-Lagrange equation

$$\frac{d}{\mathrm{d}t} \frac{\partial \mathcal{L}}{\partial \dot{x}} - \frac{\partial \mathcal{L}}{\partial x} = 0 \tag{152}$$

For a classical particle, the Lagrangian is the difference between the kinetic and the potential energy $\mathcal{L} = T - V$.

The Hamiltonian is defined as a Legendre transform of the Lagrangian

$$\mathcal{H} = \sup_{\dot{x}} \{p \cdot x - \mathcal{L}(x, \dot{x}, t)\} \tag{153}$$

The corresponding Hamilton-Jacobi equation reads

$$\partial_t S + \mathcal{H} = 0 \tag{154}$$

