# OpenReview forum: "SOLVING SCHRODINGER BRIDGE PROBLEM VIA STOCHASTIC ACTION MINIMIZATION"
_ICLR.cc/2024/Conference — Submitted to ICLR 2024_

### Official Review · Reviewer_yQV5 · 2023-10-13

**Soundness:** 3 good
**Presentation:** 2 fair
**Contribution:** 2 fair
**Rating:** 3
**Confidence:** 4

**Summary:**

The authors introduced stochastic Lagrangian formulation based on stochastic action. The authors proposed to use stochastic
variational minimization principle to train SB networks which incorporate necessary physical constraints such as Fisher information and cross-entropy. Interestingly, the governing dynamical equations are follow the Schrodinger equation with imaginary part.

**Strengths:**

a) The interpretation of Schrodinger bridge as a heat equation in imaginary time seems quite interesting to the generative model community.

b) Incorporating necessary physical constraints such as Fisher information seems promising.

**Weaknesses:**

This paper appears to be a solid paper but not ready.

1. Experiments are only evaluated on baby experiments.

2. Comparison between the proposed formulation (including imaginary part) and the original one (Tianrong's version.) is not studied enough.

3. It is a little bit ad-hoc to use the combination of Yasue Lagrangian, GM Lagrangian (19) and Lagrangian associated with cross entropy。 More principled approach is suggested.

**Questions:**

What is the benefit using the proposed Langeragian formulation?

---

### Official Review · Reviewer_qM7P · 2023-10-30

**Soundness:** 3 good
**Presentation:** 3 good
**Contribution:** 3 good
**Rating:** 5
**Confidence:** 2

**Summary:**

The paper deals with a mathematically elegant take on entropy-penalized optimal transport problems, made popular by Chen, Georgiou, and Pavon (SIAM Review 2021, https://epubs.siam.org/doi/10.1137/20M1339982). It considers the Lagrangians of Yasue (1981, https://www.sciencedirect.com/science/article/pii/0022123681900793) and Guerra and Morato (1983, https://journals.aps.org/prd/abstract/10.1103/PhysRevD.27.1774) and shows:
-- the stationary points of Yasue and Guerra-Morato actions satisfy the time-dependent
Schodinger equation.
-- it expresses the relative entropy between forward and backward probability densities, when the probability density is decomposed as forward and backward paths.

**Strengths:**

The paper deals with a mathematically elegant take on entropy-penalized optimal transport problems, made popular by Chen, Georgiou, and Pavon (SIAM Review 2021, https://epubs.siam.org/doi/10.1137/20M1339982).

The paper deals with the Lagrangians of Yasue (1981) and Guerra and Morato (1983), which should be better known.

**Weaknesses:**

Some of the theorems are phrased quite awkwardly (e.g., Theorem 4. "Stochastic action minimization of GM Lagrangian (19) is associated Fisher information production")

It's not really clear that the "Toy model" example is reproducible.

When discussing the algorithmic approaches in Section 5.1, the authors say "forward and backward loss functions (35) and (36) were iteratively minimized until convergence", but do not explain what sense of (weak) convergence they consider.

There are many typos (ßaction).

**Questions:**

Would you have some more elaborate examples of use? When submitting to "generative models" area, it would be great to explain at least in words only, what is the relevance to  "generative models".

In Section 5.1, what notion of convergence you consider in "forward and backward loss functions (35) and (36) were iteratively minimized until convergence"?

---

### Official Review · Reviewer_krSx · 2023-10-31

**Soundness:** 2 fair
**Presentation:** 2 fair
**Contribution:** 1 poor
**Rating:** 3
**Confidence:** 4

**Summary:**

The authors discuss the connection between the Schrödinger Bridge problem and stochastic mechanics. They consider the stochastic mechanics problem of minimizing the stochastic action functional with different Lagrangians. Specifically, the authors consider this functional with Yasue and Guerra-Morato Lagrangians and analyze their properties. Based on the discussed properties of Lagrangians, the authors propose to optimize stochastic action with effective Lagrangian, which is a sum of several considered Lagrangians. Furthermore, they note that it corresponds to an iterative proportional fitting procedure. Finally, the authors demonstrate their approach on toy 2D data setup.

**Strengths:**

The main strength of the paper is that it proposes to search for links between Stochastic mechanics and the Schrodinger Bridge problem. It is promising to find novel approaches to solve the Schrodinger Bridge problem, which recently became the focus of intensive study thanks to its numerous applications in generation models and analysis of biological data. Furthermore, the authors show how to use stochastic mechanics theory to derive an IPF-based algorithm to solve the Schrodinger Bridge problem.

**Weaknesses:**

*Major*

- Theorems 1, 3, and 4 are known results in the field, and the authors provide links to the corresponding articles, so only theorem 2 may be considered novel. It would be better to present the results of theorems 1,3 and 4 in the distinct background section. Otherwise, - it is unclear whether the authors claim these results as their own.
- It is not clear how theorem 2 actually relates to the Schrödinger Bridge problem.
- The proposed IPF-based algorithm and objectives for optimization are already known [1, Eq 18, 19].
- The authors evaluate the proposed approach only on the one toy 2D setup, and their approach shows poor performance even in matching the marginal distributions at initial and final times.
- The authors do not provide any experimental evidence of whether their algorithm is capable of restoring true trajectories of the Schrödinger Bridge.
- There is no discussion of previously developed methods (e.g. [1, 2, 3, 4, 5]) for the Schrödinger Bridge problem and how the proposed method relates to them.

*Minor*
- There is an error in the title (*Schrödinge* instead of *Schrödinger*).

In the end, although there are some intesting ideas, the overall paper seems to be clearly rushed and unfinished. Hence, I put “reject”.

*Links*

[1] Chen T., Liu G. H., Theodorou E. Likelihood Training of Schrödinger Bridge using Forward-Backward SDEs Theory //International Conference on Learning Representations. – 2021.

[2] Vargas, Francisco, et al. "Solving schrödinger bridges via maximum likelihood." Entropy 23.9 (2021): 1134.

[3] De Bortoli, Valentin, et al. "Diffusion Schrödinger bridge with applications to score-based generative modeling." Advances in Neural Information Processing Systems 34 (2021): 17695-17709.

[4] Gushchin, Nikita, et al. "Entropic neural optimal transport via diffusion processes." In Advances in Neural Information Processing Systems, 2023

[5] Shi, Yuyang, et al. "Diffusion Schr\" odinger Bridge Matching." In Advances in Neural Information Processing Systems, 2023

**Questions:**

- Why the optimization of the proposed effective lagrangian is equivalent to solving the Schrödinger Bridge? It seems that there should be an additional theorem that summarized the previous discussion of the properties of different Lagrangians.
- In theorem 2, it is claimed that stationary points of Yasue and Guerra-Morato Lagrangians are solutions to the time-dependent Schrodinger equation in a particular form. How does it relate to the solution to the Schrödinger Bridge problem?

---

### Official Review · Reviewer_RX7G · 2023-11-01

**Soundness:** 1 poor
**Presentation:** 1 poor
**Contribution:** 1 poor
**Rating:** 3
**Confidence:** 4

**Summary:**

The paper discusses the Schrödinger bridge problem and its application to finding optimal diffusion trajectories between probability distributions. The authors propose using stochastic Lagrangian and stochastic action as loss functions, incorporating physical constraints into the Lagrangian. They claim that this approach improves the training process and provides a more intuitive understanding of the loss function and demonstrate their result on a single example on 2D toy spirals, without any comparison to an alternative SB training formulation.

**Strengths:**

This paper takes an interesting perspective on training Schrödinger bridges using stochastic Lagrangian and stochastic action as alternatives loss functions. The authors effectively combine rich theoretical concepts in their approach to training neural network parameterized Schrödinger bridges, using theoretical underpinnings to drive advancements in practical algorithmic applications.

**Weaknesses:**

> "Our approach is the most similar to the work of (Chen et al. (2021a)) on likelihood training of the SBP."

An important and highly relevant reference was presented by Neklyudov et al. (2023). A direct comparison and discussion on how your proposal differs from that approach is necessary for a successful publication of this work.

- Neklyudov, Kirill, et al. "Action Matching: Learning Stochastic Dynamics from Samples." International Conference on Machine Learning (ICML)(2023).

> "Our approach offers great simplicity since it does not require transformation into the Euler-Lagrange equations."

There are several approaches that propose alternatives to simulation via Euler-Lagrange equation, i.e., Tong et al. (2023) and Shi et al. (2023) but also proposals by Liu et al. (2023) and Somnath et al. (2023). I would highly recommend comparing to them apart from standard SB solvers such as De Bortoli et al. (2021) and Chen et al. (2022).

- Liu, Guan-Horng, et al. "I$^ 2$SB: Image-to-Image Schrödinger Bridge." International Conference on Machine Learnint (ICML) (2023).
- Somnath, Vignesh Ram, et al. "Aligned Diffusion Schrödinger Bridges." Conference on Uncertainty in Artificial Intelligence (UAI) (2023).

> "We demonstrate the utility of our approach on the toy model dataset in 2D."

The authors propose a new training method for solving the Schrödinger bridge problem. However, the current analysis is reduced to a single experiment on a toy model dataset in 2D. And looking at the results, for anyone who ever trained a simple SB, the performance is underwhelming.
In order to fully assess the utility and potential of this method, it would be crucial to test it on more complex tasks and provide a direct comparison to existing methods that are all published alongside with code. This would provide a more comprehensive evaluation and help validate the usefulness of the proposed approach.

> "In section (2) we review the background on the SBP problem. In the following section (3) we introduce the stochastic ßaction."

The paper contains a lot of spelling errors and requires another round of proofreading, e.g., ßaction -> action, SBP problem -> SB problem, there are a lot of "the"s missing, equations (See appendix -> equations (see appendix, amsmath, etc.

Besides the language errors, the paper is citing very sparsely, e.g., for statements such as "system of equations is usually solved using two neural networks" it is always a good idea to credit the authors and provide those readers not familiar with the work with useful references to study. Concretely, citations are missing at important positions throughout the paper, and important literature is not cited in the first place.

**Questions:**

In your keywords, you mention "single-cell, trajectories". I cannot find it even mentioned in the paper. Could you point me to it?

You are claiming to provide an "intuitive grasp of the loss function". This is not clear to me after reading this paper. Could you elaborate on this claim?

---

### Official Review · Reviewer_tBew · 2023-11-05

**Soundness:** 2 fair
**Presentation:** 1 poor
**Contribution:** 2 fair
**Rating:** 3
**Confidence:** 4

**Summary:**

In this paper, he authors introduce a stochastic Lagrangian and stochastic least action principle for the solution of the Schroedinger bridge problem. The authors introduce an effective Lagrangian for creating numerical solutions for the Schroedinger bridge problem, that is optimised during training the neural network used to simulate the forward and backward flows associated with he problem.

However the authors do not discuss what is the advantage of this approach compared to existing methods for obtaining numerical solutions of the same problem, or for example the recent "Score Difference Flow"[1], neither they compare with numerical experiemnts their approach to existing frameworks that tackle the same problem.




----

**References:**

[1] Weber, Romann M. "The Score-Difference Flow for Implicit Generative Modeling." Transactions on Machine Learning Research (2023).

**Strengths:**

- the authors introduce formalism from stochastic mechanics to tackle the Schroedinger bridge problem.

**Weaknesses:**

- Unsatisfactory presentation and writing style for the conference.
- The authors do not comment on the advantages of their approach compared to existing mentioned approaches for providing numerical solutions for the same problem.
- Limited demonstration on numerical experiments (the authors show only. a single toy example)
- No comparison with competing methods.
- They provide a lot of unecessary historical information and in the Appendix they repeat widely known and available calculations, thereby obscuring their contribution.

**Questions:**

- What would be the advantage of the propopsed approach compared to the existing ones ((Chen et al. (2021a); De Bortoli et al. (2021); Shi et al. (2023); Vargas et al. (2021); Tong et al. (2023)); Weber (2023)  )?

- How does the performance and computational complexity of the method scale with system dimension?

---

### Meta-Review · Area_Chair_f6WB · 2023-12-07

**Metareview:**

The paper discusses the connection between the Schrödinger Bridge problem and stochastic mechanics.
Reviewers have agreed that despite some interesting ideas the paper is still half-baked and need some
more works before reconsideration.

**Justification For Why Not Higher Score:**

The paper has been rushed and need major reworking.

**Justification For Why Not Lower Score:**

na

---

### Decision · Program_Chairs · 2024-01-16

Reject